# Experimental evolution reveals hidden diversity in evolutionary pathways

**Peter A Lind[1,2]\*[†], Andrew D Farr[1,2], Paul B Rainey[1,2,3]**

[1]New Zealand Institute for Advanced Study, Massey University, Auckland, New Zealand; [2]Allan Wilson Centre for Molecular Ecology and Evolution, Massey University, Auckland, New Zealand; [3]Max Planck Institute for Evolutionary Biology, Plön, Germany

**Abstract** Replicate populations of natural and experimental organisms often show evidence of parallel genetic evolution, but the causes are unclear. The wrinkly spreader morph of *Pseudomonas fluorescens* arises repeatedly during experimental evolution. The mutational causes reside exclusively within three pathways. By eliminating these, 13 new mutational pathways were discovered with the newly arising WS types having fitnesses similar to those arising from the commonly passaged routes. Our findings show that parallel genetic evolution is strongly biased by constraints and we reveal the genetic bases. From such knowledge, and in instances where new phenotypes arise via gene activation, we suggest a set of principles: evolution proceeds firstly via pathways subject to negative regulation, then via promoter mutations and gene fusions, and finally via activation by intragenic gain-of-function mutations. These principles inform evolutionary forecasting and have relevance to interpreting the diverse array of mutations associated with clinically identical instances of disease in humans.

**\*For correspondence:** peter.lind@imbim.uu.se

**Present address:** [†]Department of Medical Biochemistry and Microbiology, Uppsala University, Uppsala, Sweden

**Competing interests:** The authors declare that no competing interests exist.

**Reviewing editor**: Wenying Shou, Fred Hutchinson Cancer Research Center, United States

## Introduction

Prediction of evolutionary change from a set of first principles, even in the most elementary of biological systems, has proven difficult (*de Visser and Krug, 2014*). This is due in part to the stochastic nature of mutation, but also to lack of understanding of the molecular properties of gene products and their interactions, that is, the processes underpinning development of phenotypes—including genetic and developmental constraints—which are themselves a product of the genotype-to-phenotype map (*Pigliucci, 2010*).

The ubiquity of parallel genetic evolution, observed both in nature and in laboratory experiments (*Flowers et al., 2009*; *Gerstein et al., 2012*; *Meyer et al., 2012*; *Zhen et al., 2012*; *Herron and Doebeli, 2013*; *Stern, 2013*) shows that evolution can be highly reproducible. While repeated evolution of similar traits is considered evidence of adaptive evolution, it also suggests the possibility that evolution may be governed by rules that if understood, would lead to a more predictive science (*Hansen, 2006*; *Bull and Molineux, 2008*; *Stern and Orgogozo, 2009*; *de Visser and Krug, 2014*; *Neher et al., 2014*).

Not all explanations for parallel evolution necessitate the existence of underlying rules. If evolution proceeds via a single route—because there is no other—then there is no reason to suppose that evolution is anything other than idiosyncratic (*Jost et al., 2008*; *Zhen et al., 2012*; *Vogwill et al., 2014*). Should evolution proceed along a single pathway when multiple are available and yet the fitness of the phenotype from the common path is superior, then—other than the pleasure of discovery—there is no dilemma to solve. If, however, evolution proceeds along a single pathway and yet that pathway is just one of a number of possible routes to a range of phenotypes with equivalent fitness, then determining the underlying causes becomes a matter of interest. Unfortunately

**eLife digest** Different living things often develop similar strategies to adapt to the environments in which they live. Sometimes two species that share a common ancestor independently evolve the same trait by changing the exact same genes. This is called 'parallel evolution', and it has led some scientists to ask: are there certain traits that can only evolve in a limited number of ways? Or are there other ways to evolve the same trait that, for some reason, are not explored?

Experimentally, investigating these questions is challenging, but parallel evolution occurs in the laboratory as well as in the wild. Many commonly studied organisms—such as fruit flies or bacteria—can be used in relevant studies, because they can be grown in large numbers and then exposed to identical environments. However, if this method fails to find a new way that a trait can evolve, it doesn't mean that alternative mechanisms do not exist.

Lind et al. used a different approach that instead relies on removing all of the known pathways that can be mutated to produce a given trait and then seeing if that trait can still evolve via mutations elsewhere. The experiments involved a bacterium called *Pseudomonas fluorescens* that can evolve to grow flattened and wrinkled colonies (instead of smooth, round ones) when it has to compete for access to oxygen.

Previous experiments had shown that the evolution of the so-called 'wrinkly spreader' form can be caused by mutations in one of three biological pathways. But *P. fluorescens* can survive unharmed without these pathways, which enabled Lind et al. to ask if there might be other ways that this trait could evolve. Bacteria without these three pathways were engineered and then grown under oxygen-deprived conditions. This experiment produced 91 new mutants that each had the wrinkly spreader phenotype. Further experiments revealed that together these mutants represented 13 previously unrecognized ways that the 'wrinkly spreader' phenotype can evolve.

The new rare mutants had similar fitness as the previously known, common ones—so this cannot explain why they hadn't been seen before. Lind et al. instead suggest a set of principles to explain why these newly discovered pathways are rarely mutated and how genetic constraints can bias the outcome of evolution. Further work could investigate whether these principles can help us to predict the course of evolution in other biological contexts, such as in the evolution of antibiotic resistance.

opportunities for discovery of such pathways are limited (*Gompel and Prud'homme, 2009*; *Stern, 2013*) (*Figure 1A*).

One way to proceed is via model populations amenable to replication. The outcome of evolution in each population—propagated under identical conditions—can then be determined. Discovery of a novel solution would indicate the existence of at least two pathways—moreover, the relative fitness of types can be determined. While in principle straightforward, progress requires the analysis of many thousands of replicate populations (*Desai, 2013*)—or molecules (*Ellington, 1994*)—and even if possible, failure to find an alternate route would not mean that one does not exist.

An alternative approach is to take an experimental system where the most common genetic (mutational) pathways to a particular adaptive phenotype can be identified and then eliminated without deleterious effects on fitness of the ancestral type (*Heineman et al., 2009*). Thus evolution can be re-run from different starting genotypes that differ in the spectrum of pathways available to evolution (*Figure 1B*). If the same phenotypic solution can arise from a genotype devoid of the typically used genetic route, then it is clear that evolution has more options at its disposal than are typically realized.

Precisely this approach was taken previously using experimental populations of *Pseudomonas*. *McDonald et al. (2009)* revealed the existence of three commonly used mutational routes to a single adaptive 'wrinkly spreader' (WS) phenotype, but showed that additional, less frequently utilized pathways existed (the nature of these pathways was not determined). Here, beginning with the ancestral type devoid of the three known routes to WS, we propagated multiple independent populations and identified 91 new WS mutants with similar fitness to the common WS types. A combination of genetics and genome sequencing revealed ten new single mutational step routes and three additional paths requiring two or more mutations. Our data provide an explanation for why the newly discovered pathways are rarely followed, provide a set of hierarchical principles, and show how genetic constraints can bias the outcome of evolution.

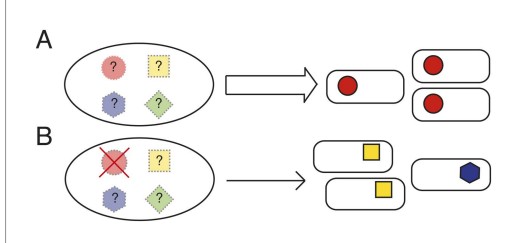

**Figure 1**. Determining the causes of parallel evolution. (**A**) A selective challenge repeatedly leads to a phenotypic adaptation with the same mutational solution represented by red circles. If only one solution is ever observed (the red circles), how can it be proven that alternative pathways exist (the unrealized [shaded] mutational pathways marked with '?')? (**B**) In an experimental system where common genetic pathways to an adaptive phenotype can be removed (the red circle with a cross), alternative pathways (blue and yellow) can be revealed and the underlying causes of the parallel evolution determined.

# Results

## Experimental rationale

The *Pseudomonas fluorescens* SBW25 (*Silby et al., 2009*), hereafter 'SBW25', experimental system of adaptive radiation has been extensively studied (*Rainey and Travisano, 1998*; *Spiers et al., 2002*; *Bantinaki et al., 2007*; *McDonald et al., 2009*) and has several features that makes it ideal for addressing the question of bias in evolutionary pathways. Every time the ancestral SM genotype of SBW25 is placed in a nutrient-rich static microcosm, metabolism-driven depletion of oxygen imposes strong selection for mutants that colonize the oxygen replete air–liquid interface (*Figure 2A*). The most successful of various mat-forming types (*Ferguson et al., 2013*) display a wrinkled morphology on agar plates (*Figure 2A*) and are known collectively as wrinkly spreaders (WS). In a previous experiment, the mutational origins of 26 independent WS genotypes were unravelled (*McDonald et al., 2009*). All mutations resided in one of three pathways (Wsp, Aws, and Mws). Each pathway harbours a di-guanylate cyclase (DGC) responsible for production of cyclic-di-GMP (c-di-GMP). When the cognate DGC is constitutively activated, cells over-produce an acetylated cellulose polymer (*Spiers et al., 2002*, *2003*)—the proximate cause of the WS phenotype (*Figure 2B*) (*McDonald et al., 2009*).

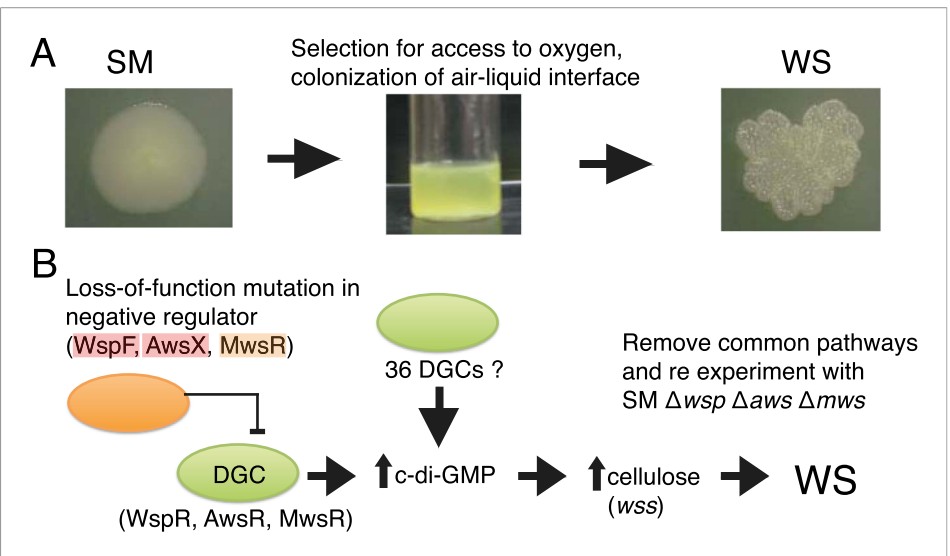

**Figure 2**. The *Pseudomonas fluorescens* SBW25 wrinkly spreader model. (**A**) The ancestral smooth (SM) strain evolves to colonize the air–liquid interface of a static microcosm. The ability to make a mat at the air–liquid interface is dependent on mutational activation of the gene products of the *wss* operon, which encodes the biosynthetic machinery for production of cellulose that function as an extracellular glue (*Spiers et al., 2003*). These mat-forming types display a wrinkled morphology on agar plates and are referred to as wrinkly spreaders (WS). Over-activation of cellulose production is caused by an increase in the second messenger c-di-GMP—the product of diguanylate cyclases (DGCs) (*Goymer et al., 2006*; *Malone et al., 2007*). (**B**) All WS mutants described in previous work have mutations in one of three loci (*wsp*, *aws*, and *mws*), all involving DGCs under negative regulation (*McDonald et al., 2009*). There are 36 additional DGCs in the genome, and the ability to remove the three commonly followed pathways makes it possible to determine whether evolution can follow alternate routes to the WS phenotype.

That evolution followed just three pathways was unexpected given that the SBW25 genome carries 39 putative DGCs (*McDonald et al., 2009*). The majority of mutations were loss-of-function changes in negative regulators of the *wsp*, *aws*, and *mws*-encoded DGCs (*McDonald et al., 2009*). Such a spectrum of mutations made sense given that loss-of-function mutations in each negative regulator resulted in constitutive activation of the DGC (*Goymer et al., 2006*; *Malone et al., 2007*), with ensuing downstream effects (*Figure 2B*) (*McDonald et al., 2009*). No evidence of mutational hotspots—a possible cause of bias—was obtained.

A unique feature of the experimental *Pseudomonas* model is the ability to remove the common pathways from the ancestral SM type without deleterious effect on fitness (*McDonald et al., 2009*). This allows a test of the hypothesis that there exist alternative evolutionary pathways to WS that are not typically followed, either because the WS types that are generated have low fitness or because properties of the genotype-to-phenotype map make alternate routes unlikely.

## Selection for new WS types and identifying mutations

200 independent glass microcosms were inoculated with the smooth (SM) Δ*wsp*Δ*aws*Δ*mws* mutant. After 6 days growth in static microcosms, dilutions were plated on KB agar and the resulting colonies screened for types exhibiting the WS morphology. WS types were found in 91 microcosms, in contrast, when the founding genotype is ancestral SM SBW25, all microcosms harbour WS types after just 3 days of propagation (*McDonald et al., 2009*).

The mutational causes of WS types were sought by suppressor analysis, using a transposon mutagenesis screen to find candidate loci for targeted Sanger sequencing (*Giddens et al., 2007*) or by genome re-sequencing. We found single mutations in 86 of the mutants in ten different loci, all encoding a protein with a putative DGC domain, similar to the three previously known pathways. Four of the remaining WS types had double mutations and one genotype had three mutations. The mutational targets are summarized in *Figure 3* and *Table 1* and full details are available in *Figure 3—source data 1*.

## Intragenic mutations

The most common of the previously unseen routes to WS (43/91) involved mutation in PFLU0085, a putative DGC lacking other annotated domains. All 22 (7 unique) base pair substitutions (BPSs) were clustered in the region 1223–1340 of the open reading frame upstream of the DGC domain. Further disruptive mutations within PFLU0085 were identified including in-frame deletions (20 mutants, 3–477 bp) and a 141-bp duplication—all in the same 1223–1340 region—suggesting that a wide variety of disruptive mutations within a small window can produce WS. This implicates 1223–1340 as a negative regulator of the downstream DGC domain.

Four other genes harboured intragenic mutations, but these were less common, suggesting a smaller target size less consistent with a negative regulatory role. One mutation (R321L) was found in PFLU0956 close to the third predicted transmembrane helix preceding the DGC domain. A similar pattern was evident at PFLU3448 where two mutations affecting the same amino acid (A200V and A200T) were identified. This residue resides close to the last of seven predicted transmembrane helices. At PFLU3571, two mutations were identified—both with W13R substitutions in the first of two transmembrane helices. A third mutation in PFLU3571 was found in the C-terminal end of the HAMP domain close to the N-terminal part of the DGC domain. The relative rarity of these mutations compared to those in PFLU0085 suggests a reduced target size that is again inconsistent with a negative regulatory role. More likely, these mutations bring about changes in protein–protein interactions, changes in localization of diguanylate cyclases in the cell, or alterations in the relative orientation of domains. All such alterations could reasonably activate production of c-di-GMP (and thus cellulose) without increasing catalytic rate. The fourth gene—that encoding PFLU5960—contained a mutation (D160G) in the DGC domain itself. The affected amino acid is close to the active site (amino acids 200–205) based on a Phyre2 structure prediction model (*Kelley and Sternberg, 2009*) of the protein (*Supplementary file 1*). Such a mutation likely increases catalytic activity of the imperfect GSDEF site.

## Promoter mutations

Mutations were found in the upstream region of three DGC-encoding genes (*Figure 3*, *Figure 3—source data 1*). The most common was associated with PFLU0956 where six BPSs and three indels in the −54 to −59 region relative to the start codon were identified. Six BPSs and one 14

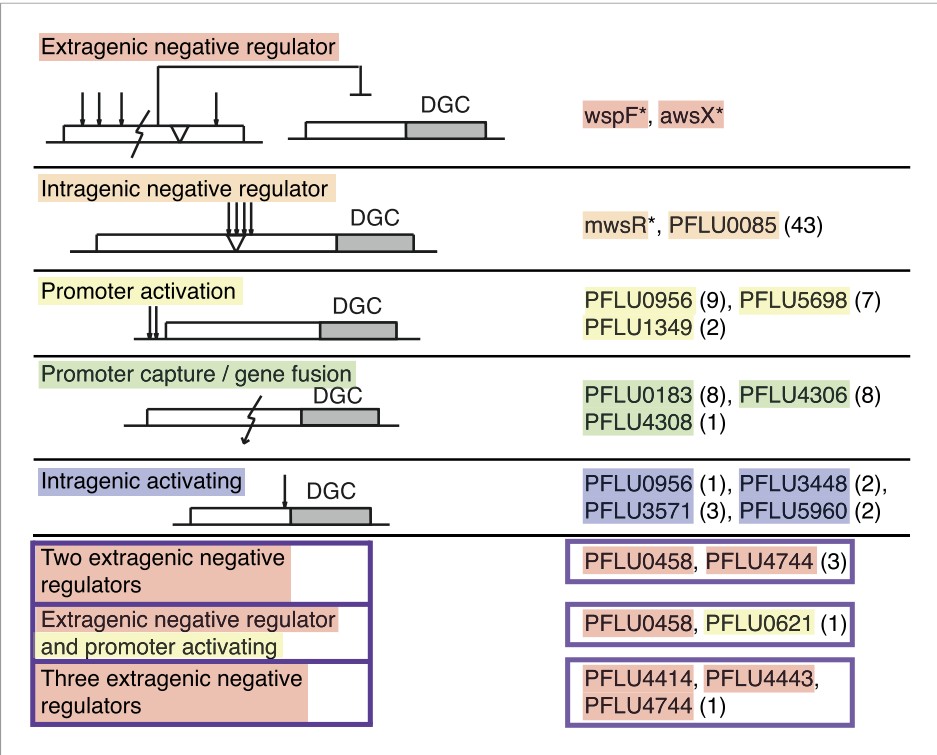

**Figure 3**. Mutational routes to WS. Numbers in parentheses are the number of independent mutants found out of 91. Putative functional effects of mutations are based on mutational patterns. Classification of mutational types has been colour coded: Red, extragenic negative regulators; orange, intragenic negative regulators; yellow, promoters (activating mutations); green, gene fusion/promoter capture; blue, intragenic (activating mutations); and purple, double and triple mutations.

The following source data is available for figure 3:

**Source data 1**. Full genetic data for all new WS mutants.

bp duplication were found upstream of PFLU5698 and two indels were found upstream of PFLU1349. In all cases, the position of the mutation indicates no impact on the ribosomal binding site. The effect of a subset of these promoter mutations on transcription is described below.

## Gene fusions

For three loci, deletion mutations caused fusions between an open reading frame encoding a DGC domain and an upstream gene. Such mutations could lead to an increase in transcription by promoter capture, or may change the functional or regulatory connections of the protein resulting in activation of the associated DGC. For PFLU0183, eight deletions were identified: each generating an in-frame fusion with a putative fatty acid desaturase (PFLU0184) located upstream. Eight deletions generating in-frame fusions were also found between PFLU4306 and PFLU4305. PFLU4306 encodes a DGC domain protein and the upstream gene encodes a putative L-lactate dehydrogenase. A third example was defined by a single mutation that fused the GGDEF domain protein PFLU4308 to PFLU4313—the latter located upstream of the DGC and encoding a hypothetical protein. This last fusion involved deletion of four intervening genes. For two of the loci WS-generating mutations arose at frequencies similar to those in promoters, suggesting that gene fusions and promoter capture play a major evolutionary role in gene and/or protein evolution.

## Double and triple mutations

Remarkably, double mutations in the same two genes, PFLU0458 and PFLU4744 were found in three independent WS mutants. Both contain early frame shift mutations or stop codons suggesting these

**Table 1**. Summary of mutations in different categories

| Category | Gene locus | Types of mutation | Number of mutants |
|---|---|---|---|
| Extragenic negative regulator | wspF (PFLU1224) | deletion, insertion, stop, frameshift, amino acid substitution | previous work* |
| | awsX (PFLU5211) | deletion, insertion, stop, frameshift, amino acid substitution | previous work* |
| Intragenic negative regulator | mwsR (PFLU5329) | deletion, insertion, amino acid substitution | previous work* |
| | PFLU0085 | deletion, insertion, amino acid substitution | 43 |
| Promoter activating | PFLU0956 | substitutions, small deletion and insertions | 9 |
| | PFLU5698 | substitutions, small insertion | 7 |
| | PFLU1349 | deletion, small insertion | 2 |
| Promoter capture/gene fusion | PFLU0183 | deletion | 8 |
| | PFLU4306 | deletion | 8 |
| | PFLU4308 | deletion | 1 |
| Intragenic activating | PFLU0956 | amino acid substitution | 1 |
| | PFLU3448 | amino acid substitution | 2 |
| | PFLU3571 | amino acid substitution | 3 |
| | PFLU5960 | amino acid substitution | 2 |
| Double and triple mutants | PFLU0458 | amino acid substitution, stop codon, frameshift | 3 |
| | PFLU4744 | amino acid substitution, frame shift | |
| | PFLU0458 | amino acid substitution | 1 |
| | PFLU0621 | substitution (promoter) | |
| | PFLU4414 | frame shift | 1 |
| | PFLU4443 | stop | |
| | PFLU4744 | amino acid substitution | |

*Mutational targets described in **McDonald et al. (2009)**.

are loss-of-function mutations (**Figure 3**, **Figure 3—source data 1**). PFLU0458 harbours a DGC domain but without a catalytic motif (suggesting loss of enzymatic activity) and an EAL domain typically involved in c-di-GMP degradation. Loss-of-function mutations suggest a role for PFLU0458 as a c-di-GMP degrading enzyme, which is also supported by data from *Pseudomonas aeruginosa*, where the orthologue PA5017 (*dipA*, *pch*) lacks DGC activity, but is proficient for phosphodiesterase activity (**Roy et al., 2012**). PFLU4744 encodes the alginate biosynthesis transcriptional activator *algZ/amrZ* that has a previously known role in cellulose expression and biofilm formation (**Giddens et al., 2007**). The remaining double mutant harboured a mutation in PFLU0458, as above, and a second mutation upstream (−51) of the DGC-encoding protein PFLU0621.

The single triple mutant harboured a mutation in PFLU4744 that was found in combination with PFLU0458 described above, but in this instance was combined with mutations in two other regulatory proteins: PFLU4414 (*cheA*, chemotaxis histidine kinase) and PFLU4443 (*adnA/fleQ*, flagella activator, negative regulator of *algR*). A connection between PFLU4744/PFLU4443 and the cellulose biosynthetic (*wss*) operon is known (**Giddens et al., 2007**).

## Reconstruction of mutations and phenotypic characterization

Representative intragenic mutations, promoter mutations, and gene fusions (**Figure 4**) were reconstructed in the ancestral SM SBW25 background to confirm that they are the sole cause of the WS phenotype and also to rule out any dependency on the Δ*wsp*Δ*aws*Δ*mws* genetic background.

To confirm that the phenotypic basis of the selective advantage is similar to the previously described WS mutants (where activation of a DGC results in overproduction of cellulose [**McDonald et al., 2009**])

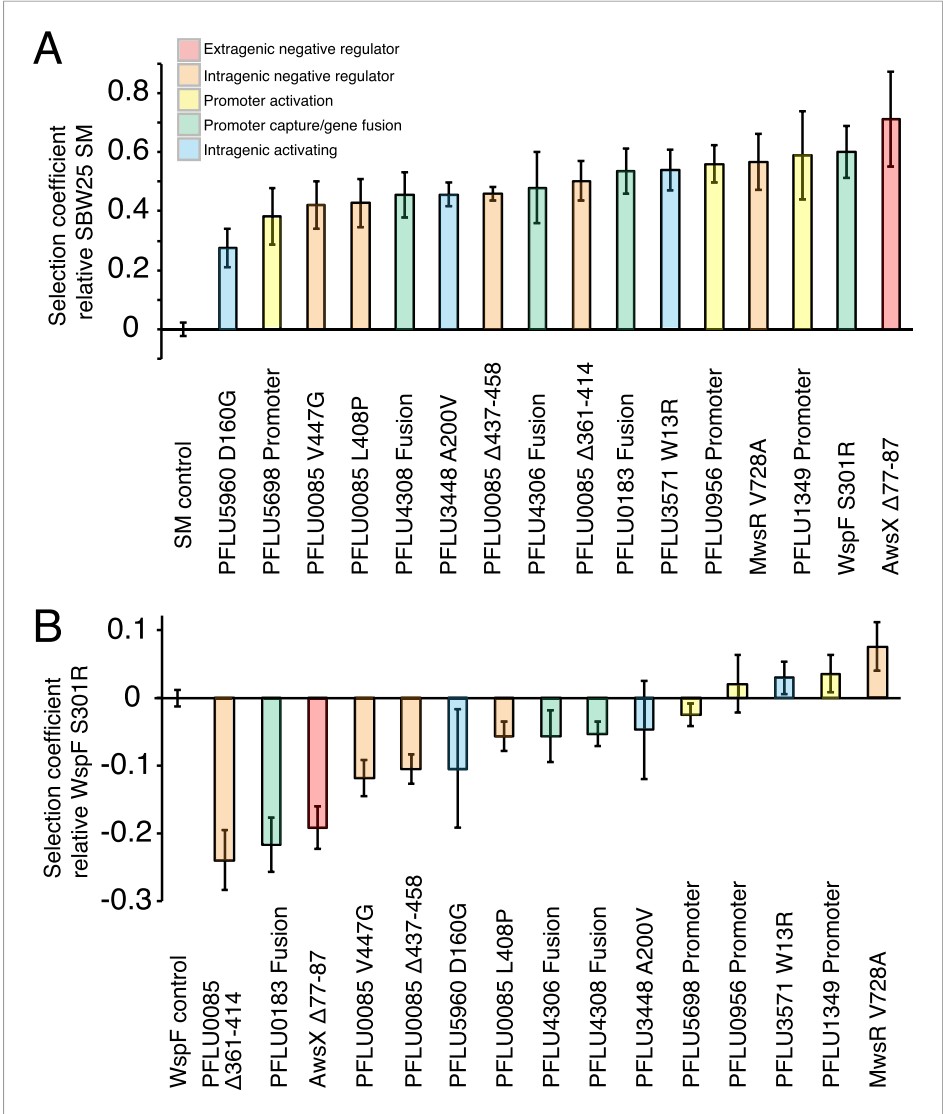

**Figure 4**. Fitness effects of WS types arising from single mutations. (**A**) Fitness in a 1:100 invasion assay against the ancestral SM genotypes. (**B**) Fitness in a 1:1 competitive assay against the high fitness LSWS (WspF S301R) genotype. Error bars represent SD (n = 8).

The following figure supplements are available for figure 4:

**Figure supplement 1**. Colony morphology of WS types arising from single mutations on KB agar with and without Congo Red.

**Figure supplement 2**. Microcosms of WS types arising from single mutations after 24 hr and 72 hr static growth.

reconstructed mutants were stained with calcofluor and Congo red and their colony morphology examined. All single mutants were positive for calcofluor staining and colony morphologies with Congo red staining showed that all stained to a higher degree than the SM ancestor, but there were differences (*Figure 4—figure supplement 1*). All reconstructed mutants colonized the air–liquid interface of static microcosms within 24 hr (*Figure 4—figure supplement 2*).

Mutations found in the double and triple mutants (*Figure 5*) were reconstructed individually and then combined. For the double and triple mutants, only combinations of the individual mutations with a PFLU4744 mutation were positive for calcofluor and Congo red binding (*Figure 5—figure supplement 1*).

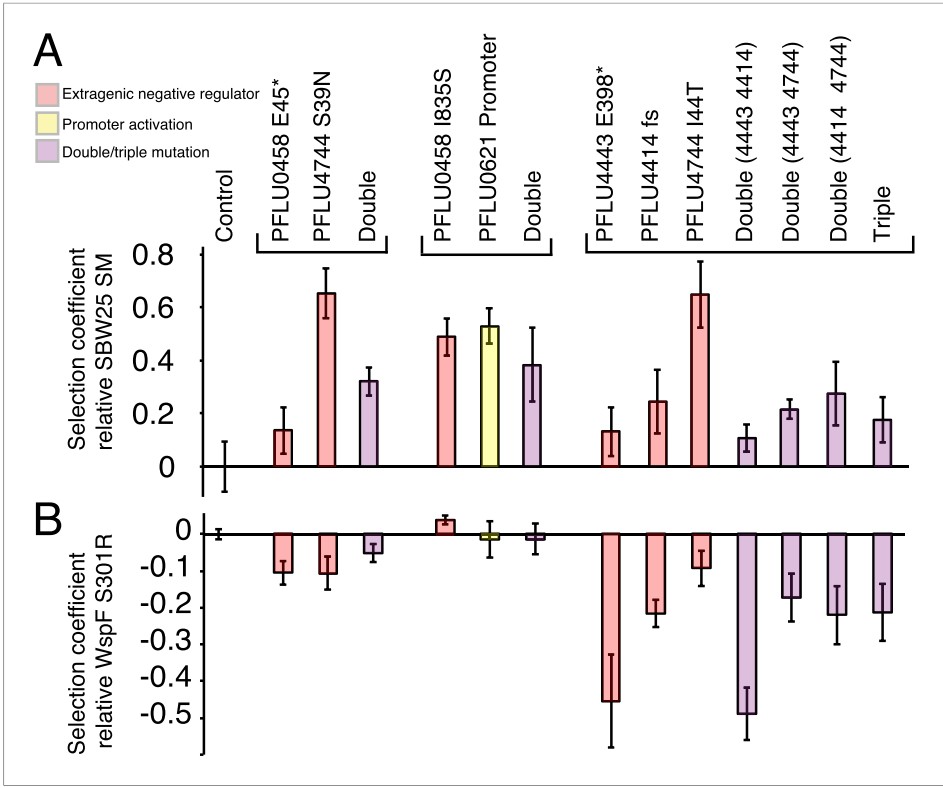

**Figure 5**. Fitness effects of WS types arising from two and three mutational events. Individual mutations were recreated in the ancestral SM genotype plus combinations of mutations (see text). (**A**) Fitness in a 1:100 invasion assay against the ancestral SM genotypes. (**B**) Fitness in a 1:1 competitive assay against the high fitness LSWS (WspF S301R) genotype. Error bars represent SD (n = 8).

The following figure supplements are available for figure 5:

**Figure supplement 1**. Colony morphology of WS types arising from two and three mutational events on KB agar with and without Congo Red.

**Figure supplement 2**. Microcosms of WS types arising from two and three mutational events after 24 hr and 72 hr static growth.

All individual and combined reconstructed mutants colonized the air–liquid interface of static microcosms within 24 hr with the exception of the genotype carrying solely the PFLU4443 E398* mutation (*Figure 5—figure supplement 2*).

## Fitness of single WS mutants

The preferential use of the Wsp, Aws, and Mws pathways to generate WS types when starting from the ancestral SM genotype could be a consequence of the WS types arising via the newly discovered pathways having lower fitness compared to those arising from mutation in the three known pathways. To investigate if reduced fitness explained the data, two types of competitive fitness assays were performed using the reconstructed mutants.

The first assay focused on the capacity of the WS types to invade, from rare, a numerically dominant population of ancestral SM types marked with GFP. This mimics the situation where a random WS mutant occurs in a population and rises to high frequency without clonal interference from other WS mutants. The second assay involved a 1:1 competitive fitness assay in which each reconstructed WS type was directly pitted against one of the most fit and common of the WS mutants previously described—the so named LSWS type (WspF S301R)—also marked with GFP (*McDonald et al., 2009*). This assay focused on competitive performance at the air–liquid interface—a realistic

situation given that several WS mutants are likely to be present in the large populations ($\approx 10^{10}$ cells) used in the evolution experiments. Thus the two assays focus on different aspects of the adaptive radiation that are not necessarily directly correlated, for example, a mutant can be superior in initial attachment, but grow slowly at the air–liquid interface. In addition, fitness in static microcosms is frequency dependent (*Rainey and Travisano, 1998*).

All WS types harbouring single mutations invaded the SM population from rare with selection coefficients (s) of 0.28–0.59 per generation compared to the control SM type (s = 0) (*Figure 4A*). Three WS mutants caused by mutations in the commonly followed mutation pathways (*wspF*, *awsX*, and *mwsR*) were included to see if lower invasion fitness was responsible for the rarity of the newly discovered mutational routes. All three produced high selection coefficients. Two newly discovered WS types containing mutations in PFLU5698 and PFLU5960 were significantly less fit than the LSWS (p < 0.01, two tailed t-test). However, taken together, the low (in a few instances) invasion rate for WS generated by mutations in the newly discovered pathways cannot account for the rarity of the alternative WS types found here.

In competition with the high fitness common LSWS (*wspF* S301R) mutant (s = 0), seven of the 13 reconstructed WS types had significantly lower fitness (p < 0.01, two tailed t-test), including all four of the PFLU0085 mutants (*Figure 4B*). One WS type harbouring a mutation in PFLU1349 had significantly higher fitness (s = 0.039, p < 0.01, two tailed t-test).

## Fitness of double and triple mutants

In the invasion assay all mutants, except the PFLU0458 E45*, PFLU4443 E398*, and PFLU4443 PFLU4414, invaded the SM population successfully (*Figure 5A*) (two tailed t-test p > 0.01 vs the control). In competition assay, the fitness of WS types harbouring two mutations were no different to WS types arising from a single mutation and were overall indistinguishable from the LSWS control strain (*Figure 5B*). The individual PFLU0458 I835S mutant has a significantly higher fitness in the competition assay (s = 0.039, two tailed t-test p = $1.8 \times 10^{-4}$). The triple mutants and all combination of genotypes carrying constituent mutations had lower fitness than LSWS in the competition assay, with the highest fitness recorded for the single PFLU4744 I44T mutant (*Figure 5B*). A similar finding came from the invasion assay where this mutant also showed high competitive ability. WS types with various combinations of single mutations ranked at the lower end of the fitness spectrum with the PFLU4443, PFLU4443/PFLU4414, and triple mutation WS types being no different from the SM control (two tailed t-test p > 0.01, *Figure 5A*).

## Transcriptional effects of promoter mutations and gene fusions

WS types carrying putative promoter mutations (PFLU0956, PFLU1349, PFLU5698, and PFLU0621) were expected to activate DGCs by increasing transcription. This is a possible mechanism of activation for the gene fusions (PFLU0183, PFLU4306, and PFLU4308) where new promoters may have been captured by the deletion event or may have resulted in transcriptional terminators being lost. To examine the effect on transcription of representative promoter and gene fusion mutants, we performed quantitative PCR. The promoter mutations all increased transcription, up to 40-fold compared to SBW25 (*Figure 6*): the PFLU0621 mutant showed a more modest fivefold increase. The latter result is consistent with the fact that this mutation alone does not generate the WS morphology (*Figure 5—figure supplement 1*). WS types carrying gene fusion mutations (PFLU4305-4306 and PFLU4313-4308) also showed large increases in transcription with 23- and 219-fold increases respectively, supporting promoter capture as the mechanism of activation. The PFLU0184-0183 fusion showed a 3.9-fold increase in transcription, which is not significantly different (p = 0.09, two tailed t-test) to a control PFLU0085 mutation (*Figure 6*).

## Discussion

Understanding genetic evolution requires knowledge of the factors that affect the translation of mutation into phenotypic variation. While much is known about the nature of mutation (*Drake et al., 1998*), knowledge of how change in DNA sequence is translated into phenotypic variation—the raw material for natural selection—is less well understood (*Gompel and Prud'homme, 2009*). Of particular relevance is the complex network of functional and regulatory connectivities that define the genotype-to-phenotype map. This network of interactions constrains (channels) evolution; restricts the pathways it takes and—by imposing limits to phenotype space—defines the rules by which it works

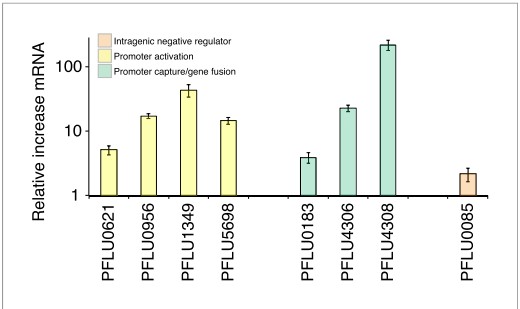

**Figure 6**. Relative increase of mRNA levels in WS mutants with putative promoter or gene fusion mutations compared to SM SBW25 measured by quantitative PCR. PFLU0085 is the Δ361–414 deletion mutant not expected to result in increased mRNA levels used as a control. Error bars represent SEM.

(*Hansen, 2006*; *Gompel and Prud'homme, 2009*; *Stern and Orgogozo, 2009*). The notion that evolution follows a limited subset of pathways and conforms to rules is not new (*Geoffroy Saint-Hilaire, 1818*; *Vavilov, 1922*; *Smith et al., 1985*), however, its relevance has often been challenged, largely through too frequent invocation of constraints to explain negative data and through lack of experimental insight (*Brakefield and Roskam, 2006*).

Our unique experimental system has allowed unveiling of hitherto hidden evolutionary pathways by which the WS phenotype can be achieved. For the most part, WS types arising via the new pathways do not differ in fitness relative to known WS types and therefore their rarity cannot be attributed to a selective disadvantage. Genetic constraints, however, provide plausible explanations—certain pathways have a greater capacity than others to translate mutation into phenotypic variation (*Figure 7*). While bias is well known to arise as a consequence of localized mutational hotspots, the spectrum of mutations underpinning different routes to WS suggests a minor influence of any such bias (*Figure 3—source data 1* [*McDonald et al., 2011*]). Conceivably, the presence of a mutational hot spot at the exact location of an intragenic activating mutational site or promoter could have a major impact on the rate of phenotypic production from that locus, but the diversity of mutants obtained for these mutational types suggest that such biases are not dominant in this system.

All newly revealed mutational routes to the WS phenotype harbour proteins predicted to encode DGCs, all show the distinctive WS colony morphology and niche preference, and all over-produce cellulose. Despite underlying molecular similarity only three are routinely travelled: Wsp, Aws, and Mws. The newly discovered mutational spectra allow proposals regarding probable mechanisms for activation of the pathways specific to each DGC and with this a more complete understanding of why genetic evolution prefers certain pathways over others.

From a purely genetic perspective there are no surprises: loss-of-function mutations are vastly more common that gain-of-function mutations (*Gompel and Prud'homme, 2009*; *Lee et al., 2012*; *Herron and Doebeli, 2013*) and thus those pathways containing DGCs subject to negative regulation will, by virtue of target size (a product of length of DNA and function), translate mutation into WS variation more efficiently than those pathways containing DGCs subject to other forms of regulation. But our detailed analysis makes possible a finer scale of resolution.

In common with the previously known Wsp, Aws, and Mws pathways, PFLU0085 also appears—on the basis of the spectrum of DGC-activating mutations—to be subject to negative regulation. That WS arising via the PLU0085 are not detected when the founding genotype is ancestral SM is readily understood based on the small mutational target (∼117 nucleotides) and the fact that mutations within this region cannot disrupt the reading frame else DGC activity be lost.

The second most common set of routes to WS, with 5–40 fold fewer mutations per gene compared to PFLU0085, are those involving promoter mutations and gene fusions (*Figure 3*). It is not clear why promoter mutations were only found for a minority of the 39 DGC domain-containing proteins. Possibly this reflects the need for mutations to generate a high level of transcription, but it is also consistent with the fact that regulation of many DGC-containing proteins is post-translational (*Goymer et al., 2006*; *Jenal and Malone, 2006*).

The high frequency of gene fusions is surprising given the relatively few reports of this kind of mutational event in experimental evolution studies, with the notable exception of promoter capture that was central to the evolution of an *Escherichia coli* mutant that gained the ability to grow on citrate (*Blount et al., 2012*). The loss of genetic material through beneficial gene fusions could contribute to deletional bias over evolutionary timescales as observed in bacteria (*Mira et al., 2001*) and presents a selectionist alternative to reductive evolution by genetic drift or loss of biosynthetically expensive genes (*Moran, 2002*; *Lee and Marx, 2012*). Moreover, beneficial promoter capture events can create

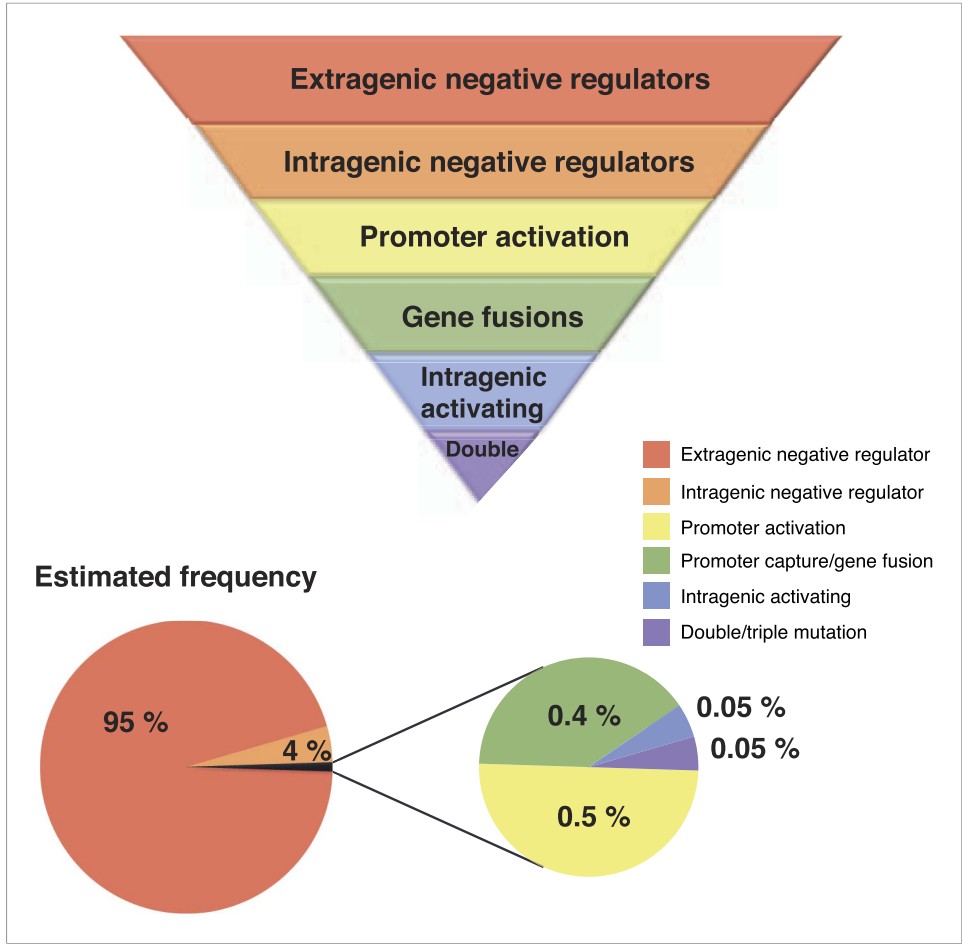

**Figure 7**. Hierarchical principles of mutational gene activation. Certain pathways have a greater capacity than others to translate mutation into phenotypic variation depending on the functional and regulatory interactions involved. The estimated number of mutants is a first approximation of how mutations are expected to be distributed between different categories assuming a similar number of target genes in each category. This distribution can also be biased by mutational hot spots and differences in fitness effects of mutations in the gene involved.

new protein domain combinations and provide raw material for the evolution of genes with novel functions, even if initially selected only because of a difference in transcriptional regulation.

The third most common mutation route to WS involves rare activating mutations that differ from intragenic negative regulators in that only specific base pair substitutions can lead to activation (*Figure 3*). Effects at the molecular level are unknown, but it appears that mutations modifying the active site are rare, given that only one mutation was found in proximity to the catalytic site. Assuming that any of the newly discovered pathways are used at a frequency that is at least 10-fold lower than for the common pathways to WS, it is possible to estimate that only one in one thousand activating mutations would directly modify the active site (*Figure 7*). This strongly supports the theory that the majority of advantageous mutations are likely to occur in regulatory regions, including promoters and regulatory proteins, or in the case of intragenic mutations, primarily in regions peripheral to the main catalytic domain (*McAdams et al., 2004*; *Wray, 2007*).

WS types arising as a consequence of two mutations arose at a similar frequency as the rare activating mutations. Such double (and triple) mutants are unexpected given the seeming improbability of such events based on estimates of the genome mutation rate (*Drake et al., 1998*). Nonetheless their detection is not uncommon (*Drake, 2007*) and they have been explained by the existence of transient phenotypic mutators in the population due to transcriptional or translational errors in the expression of key genes involved in replication or DNA repair (*Ninio, 1991*; *Drake, 2007*).

This suggests that once the third most common pathways involving single mutations have been realized, the same evolutionary principles can be applied to double mutations starting with double mutations in negative regulators, followed by double mutations in one negative regulator and one promoter mutation (*Figure 3*, *Table 1*).

Taken together our findings provide the clearest evidence yet that the network of regulatory interactions and connectivities that define the genotype-to-phenotype map directly affects the translation of mutation into phenotypic variation and that this can profoundly bias the course of genetic evolution. This has a number of implications. Firstly, our findings contribute to the growing number of studies that show that parallel phenotypic evolution is often underpinned by parallel genetic changes (see [*Stern, 2013*] *Table 1* for a summary) (*Jost et al., 2008*; *Blount et al., 2012*; *Gerstein et al., 2012*; *Meyer et al., 2012*; *Zhen et al., 2012*; *Herron and Doebeli, 2013*). However, whereas it is common to attribute such genetic parallelism to selection, our work suggests the need for caution: genetic architecture can be a significant contributory factor. It is even conceivable, within the bounds of population parameters, such as population size and mutation rate, that certain high fitness phenotypes are never realized because of genetic bias.

Secondly, the WS-based experimental system has allowed discovery of pathways that evolution would rarely ever follow and suggests a hierarchical set of rules. These rules are consistent with the concept of 'target size' (*Hansen, 2006*; *Gompel and Prud'homme, 2009*), which can be formulated more specifically in terms of size of the gene and the likelihood that changes generate viable phenotypes. The latter depends on the opportunity for loss-of-function mutations to generate adaptive phenotypes, which, in turn, depends on the function of the gene. But it is clear that while evolution will proceed most readily via loss-of-function mutations (where possible), other kinds of genes that afford a much reduced target size cannot be overlooked. As we demonstrate, there exists opportunity for promoter mutations, gene fusions, and activating mutations to also contribute to new phenotypes, but with a greatly reduced likelihood.

The existence of multiple genetic routes to a particular phenotypic end point is, given growing evidence of redundancy and evolvability in regulatory systems, possibly more common than currently appreciated (*Gompel and Prud'homme, 2009*; *Heineman et al., 2009*; *Stern, 2013*). If so, then the kinds of mutational patterns unraveled here may be evident elsewhere. We cautiously suggest that in addition to relevance for interpreting patterns of mutation underlying numerous studies of parallel and convergent evolution in natural and laboratory populations (*Jost et al., 2008*; *Blount et al., 2012*; *Gerstein et al., 2012*; *Meyer et al., 2012*; *Zhen et al., 2012*; *Herron and Doebeli, 2013*; *Stern, 2013*), our findings may also have value in understanding the spectrum of rare and common genetic variants underlying specific human diseases (*Gibson, 2011*). In the context of cancer, the mutational heterogeneity of different cancer types is well documented and it is possible that the common (and less common) mutational trajectories (*Yates and Campbell, 2012*; *Lawrence et al., 2014*) make sense in light of constraints due to genetic architecture.

We have made little of our findings in terms of the regulatory networks underpinning c-di-GMP synthesis and cellulose expression in bacteria—including mechanisms of DGC activation—but the degree of redundancy (under one set of laboratory conditions) and capacity for evolutionary change by recruitment of different DGC-containing proteins and pathways is remarkable. The evident flexibility suggests that evolutionary rearrangement of interacting modules likely occurs in natural populations—it also underpins recent experimental findings on the evolution of multicellular life cycles (*Hammerschmidt et al., 2014*). Indeed, such facility for re-wiring may partly explain the diverse modular arrangement of both DGC domain-encoding genes and DGC-containing operons found among even closely related strains of a single species (*Jenal and Malone, 2006*).

The hierarchical rules for the evolution of new WS phenotypes revealed through the analysis of experimental *Pseudomonas* populations may be sufficiently general for them to be applied to other biological settings where an adaptive challenge can be solved by mutational gene activation. Such settings include the evolution of antibiotic resistance, the evolution of virulence in pathogens and the emergence of complex traits in eukaryotic populations. However, the relevance of these principles remains to be tested. This could occur via a priori predictions of mutational routes for specific populations in given selective contexts. If robust, then the claim to have moved closer to a more predictive theory of genetic evolution will have substance.

# Materials and methods

## Strains and media

All strains used are *P. fluorescens* SBW25 (*Silby et al., 2009*) or derivatives thereof except for *E. coli* strains used for strain construction and transposon mutagenesis (*E. coli* DH5-α λ$_{pir}$, *E. coli* SM10 λ$_{pir}$ IS-Ω-kan/hah, *E. coli* pRK2013). *P. fluorescens* strains were grown in King's medium B (KB) (*King et al., 1954*) at 28°C and *E. coli* strains were grown in lysogeny broth (LB) (*Bertani, 1951*) at 37°C. Solid media were KB or LB with 1.5% agar.

Antibiotic concentrations for strain construction and plasmid maintenance were gentamycin (10 mg/l), kanamycin (100 mg/l), tetracycline (15 mg/l), nitrofurantoin (100 mg/l), and cycloserine (1000 mg/l). X-gal (5-bromo-4-chloro-3-indolyl-β-D-galactopyranoside) was used at a concentration of 40 mg/l in agar plates. Calcofluor (Fluorescent brightener 28) was added to agar plates at a concentration of 35 mg/l and Congo red at 10 mg/l.

## Experimental evolution to select for WS

Individual colonies of the smooth ancestor strain PBR716 (Δwsp Δaws Δmws) (*McDonald et al., 2009*) were used to inoculate 200 glass microcosms (6 ml KB) that were incubated statically for 72 hr at 28°C. The microcosms were then vortexed vigorously and diluted 1000 times into new KB microcosms that were incubated for another 72 hr under identical condition before suitable dilutions were spread onto agar plates. After incubation for 48 hr, the plates were screened (500–2000 colonies) for colonies with different morphology than the SM ancestor.

## Transposon mutagenesis and sequencing of transposon insertion sites

Transposon mutagenesis was used to find candidate genes for alternative WS mutations as previously described (*Giddens et al., 2007*). Briefly, the plasmid pCM639 containing the IS-Ω-kan/hah transposon was conjugated from *E. coli* SM10 λ$_{pir}$ into the recipient *P. fluorescens* WS strain using an *E. coli* pRK2013 helper strain. Suitable dilutions of successful transconjugants were selected on KB plates with kamamycin for selection of the transposon and nitrofurantoin for counterselection of *E. coli*. Fewer than 1000 transconjugants from each independent conjugation were screened for loss of the WS colony morphology and single colonies were isolated on agar plates. The insertion sites in the genome were found by an arbitrarily primed PCR approach and Sanger sequencing (Macrogen, South Korea) of the products (*Manoil, 2000*).

## DNA sequencing to find alternative WS mutations

Candidate genes from the transposon suppressor analysis were sequenced by Sanger sequencing (Macrogen) in all alternative WS in an iterative fashion, eliminating the common pathways before moving on to the next round of transposon mutagenesis and sequencing. For a few mutants that consistently failed to produce suppressors, were difficult to phenotypically distinguish from the ancestor or had low conjugation efficiency; we used genome sequencing to find the mutations. Genomic DNA was prepared using the Wizard Genomic DNA purification kit (Promega), sequenced by the Australian Genome Research Facility using Illumina HiSeq2000 and assembled against the reference *P. fluorescens* SBW25 genome using Geneious 5.5.6 (Biomatters). All oligonucleotide primers used in this study are available in *Supplementary file 2*.

## Reconstruction of mutations in SBW25

Representative intragenic mutations (PFLU0085 L408P, V447G, ΔR361-R414, ΔR437-A458; PFLU3448 A200T; PFLU3571 W13R; PFLU5960 D160G), promoter mutations (PFLU0956 T-54G; PFLU1349 ins TC -47/48, PFLU5698 C-73T), and gene fusions (PFLU0184 M1-T328 fused to PFLU0183 A29-G335; PFLU4305 M1-Y340 fused to PFLU4306 S21-G489; PFLU4313 M1-F115 fused to PFLU4308 A189-R820) were reconstructed in the SBW25 background to prove that they are the cause of the WS phenotype and to demonstrate that these mutational pathways are available in the ancestral strain and not dependent on deletion of the *wsp*, *aws* and *mws* loci. Mutations found in the double and triple mutants (PFLU0458 I835S, PFLU0621 -51C > T; PFLU0458 E45*, PFLU4744 S39N; PFLU4414 Y652fs, PFLU4443 E398*, PFLU4744 I44T) were reconstructed individually and then combined. We used a two-step allelic replacement method to transfer the mutations into the ancestral background

as previously described (*Rainey, 1999*; *Bantinaki et al., 2007*). In summary, PCR (Phusion High-Fidelity DNA polymerase, Thermo Scientific) was used to amplify an approximately 1000-bp region surrounding each mutation, and the product was subsequently cloned into the pCR8 plasmid and sequenced. The cloned fragment from pCR8 was then moved to the pUIC3 suicide vector and mobilized into *P. fluorescens* SBW25 where it integrates into the chromosome by homologous recombination. After non-selective growth in KB, 10 mg/l tetracycline was added to inhibit the growth of cells that had lost the pUIC3 insert. After 2 hr 1000 mg/ml cycloserine was added to kill growing cells and enrich for cells that had lost pUIC3 with the tetracycline resistance marker and incubated for 4 hr at 28°C. Suitable dilutions were then plated on agar plates with X-gal to allow screening for loss of the *lacZ* gene on pUIC3. White colonies were confirmed to be tetracycline sensitive and single colonies were isolated. The region containing the desired mutation was then sequenced in a number of colonies, with both the ancestral and wrinkly phenotype to exclude picking bias and confirm that the wrinkly phenotype was linked to the mutation.

## Construction of competition strains

We used the wild-type SBW25 and a previously described high fitness WS mutant LSWS (*wspF* A901C, S301R) (*Bantinaki et al., 2007*; *McDonald et al., 2009*) to construct Green Fluorescent Protein (GFP) expressing strains for use in competition experiments to determine the relative fitness of the alternative WS. These strains were genetically tagged in the chromosome with a mini-Tn7 transposon expressing GFP and a gentamicin resistance marker (miniTn7(Gm)P$_{rrnB}$ P1 gfp-a) (*Lambertsen et al., 2004*) that was transferred from *E. coli* by conjugation together with the pUX-BF13 plasmid carrying the transposase genes.

## Competition fitness assay

Competitive fitness was determined relative to the LSWS strain (*Spiers et al., 2002*; *Goymer et al., 2006*) (SBW25 *wspF* A901C, S301R) marked with GFP. Strains were grown for 16 hr, shaking, in KB at 28°C before they were mixed at equal volumes diluted 6 times and grown for 4 hr at the same conditions to ensure that the strains were in the same physiological state before the competition started. The initial ratios of alternative WS to LSWS GFP were determined by counting 100,000 cells using flow cytometry (BD FACS Canto) detecting GFP fluorescence on the 488 nm laser with 530/30 bandwidth filter. Suitable dilutions of the initial population were plated on KB agar plates to determine viable counts. The mix of alternative WS and LSWS was diluted 1000-fold in KB and incubated for 24 hr, static at 28°C. Final viable counts and ratios were determined in the same way. After incubation for approximately 40 hr at 28°C, viable counts were determined and the stability of the WS phenotypes and GFP marker was determined. This was possible as most WS types have slightly different colony morphologies, which allow us to determine if the GFP marker was lost and the emergence of smooth ancestral types is easily detected. Rarely, smooth colonies were found and if they made up more than 5% of the total population, the competition data were discarded for that microcosm. The number of generations was determined by ln(final population/initial population)/ln(2). Selection coefficients were calculated using the regression model s = [ln(R(t)/R(0))]/[t], as previously described (*Dykhuizen, 1990*) where R is the ratio of alternative WS mutant to LSWS GFP and t is the number of generations. Control experiments with LSWS vs LSWS GFP were performed to compensate for the fitness cost (s = 0.06 ± 0.01) of the miniTn7 with the GFP marker. For each strain, the competition assay was performed in quadruplicates at a minimum of two separate occasions.

## Invasion fitness assay

Invasion fitness was measured relative to the smooth (SM) ancestral SBW25 mini-Tn7 GFP. Invasion strains were grown in KB for 24 hr shaking and then mixed 1:100 with the SM GFP strain and a 1000-fold dilution of this mix was used to inoculate 6 ml static microcosms. After 48 hr at 28°C, the ratio of unmarked WS to GFP marked SM was determined by flow cytometry as described above. Viable counts on KB plates of initial and final populations were performed to calculate the number of generations during the invasion growth. The stabilities of the GFP marker and colony morphologies were confirmed and data from microcosms with >5% wrinkly GFP or smooth unmarked were discarded. Selection coefficients relative to the ancestral SBW25 strain were calculated as described above with compensation for the cost of the GFP marker determined by control invasion experiments

with SBW25 vs SBW25 GFP. The averages presented are the result of at least two independent experiments with quadruplicates for each strain.

## Quantitative RT-PCR

Total RNA was isolated from reconstructed WS mutants with mutations in upstream promoter regions or fusions to other open reading frames using the SV Total RNA Isolation system (Promega). Cells were harvested at $OD_{600} = 0.6$ and resuspended in Tris-1mM EDTA buffer with 0.4 mg/ml lysozyme to lyse the cells before using the supplied protocol. The RNA was reverse transcribed into cDNA using the High Capacity cDNA reverse transcription kit (Applied Biosystems) and diluted 40 times before use in quantitative real-time PCR (DyNAmo Colorflash SYBR Green qPCR kit [Thermo Scientific], PikoReal 96 Real-Time PCR System [Thermo Scientific]). Relative changes in mRNA levels between the ancestral SBW25 strain and the WS mutants were determined with the $\Delta\Delta Cq$ method using *recA* as an internal control (*Livak and Schmittgen, 2001*). We used two biological replicates per strain and three technical replicates per run over at least two separate experiments.

## Acknowledgements

Supported by the Marsden Fund Council from New Zealand Government funding, administered by the Royal Society of New Zealand.

## Additional information

### Funding

| Funder | Grant reference | Author |
|---|---|---|
| Royal Society of New Zealand | Marsden Fund Council | Paul B Rainey |

The funder had no role in study design, data collection and interpretation, or the decision to submit the work for publication.

### Author contributions

PAL, Conception and design, Acquisition of data, Analysis and interpretation of data, Drafting or revising the article; ADF, Conception and design, Acquisition of data, Analysis and interpretation of data; PBR, Conception and design, Analysis and interpretation of data, Drafting or revising the article

### Author ORCIDs

Peter A Lind, http://orcid.org/0000-0003-1510-8324

## Additional files

### Supplementary files

• Supplementary file 1. Phyre2 pdb model of PFLU5960.

• Supplementary file 2. Oligonucleotide primers.

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
