## [Decision Letter]

Thank you for sending your work entitled “Experimental evolution reveals hidden diversity in evolutionary pathways” for consideration at *eLife*. Your article has been favorably evaluated by Detlef Weigel (Senior editor) and three reviewers, one of whom is a member of our Board of Reviewing Editors. One of the three peer reviewers, Jim Bull, has agreed to share his identity.

The Reviewing editor and the other reviewers discussed their comments before we reached this decision, and the Reviewing editor has assembled the following comments to help you prepare a revised submission.

Lind et al. use the *Pseudomonas fluorescens* unshaken microcosm experimental evolution system to deeply profile different mutations that can give rise to the wrinkly spreader phenotype. Neither the trick of knocking out one pathway to see how evolution can still achieve a given phenotype nor the target-size argument for predicting evolutionary pathways are especially new. For example, the Barry Hall studies of evolved beta galactosidase used the former approach. There are also multiple examples of profiling the steps in evolving new metabolic capabilities in the 1984 Mortlock “Microorganisms as model systems for studying evolution” volume that follow the ‘target-size’ paradigm of a regulatory mutation knocking out a repressor first to achieve overexpression, followed by rarer refinement mutations at specific sites in an enzyme that change its specificity.

The strength of this study is in the depth with which the authors have been able to find so many different solutions to the problem of activating cyclic-di-GMP production to achieve the wrinkly spreader phenotype. It's quite interesting that there are so many different di-guanylate cyclases that can be co-opted with just a single mutation, in so many different ways. They also present nice data showing that some newly revealed WS mutations confer similar fitness advantages to the normally dominant versions, and that the putative promoter mutations do increase mRNA levels (sometimes by extremely large amounts). There are many types of evolutionary pathways revealed here, and the text also interprets them for the reader e.g., the type of mutation(s) involved and whether they destroy a function or create a new function, and what that function is. The literature on experimental evolution is actually pretty poor about noting the difference between evolution of new or altered function, versus evolution to destroy a function, but this paper makes the distinction clear; that is just one of its many attributes. Another is the fact that the selection leads to a single phenotype with a clear adaptive interpretation; many studies of experimental evolution have described the DNA bases of favored changes yet fail to connect them to a clear beneficial phenotype.

Major comments:

1) I am not sure about the “general principles” quoted in the Abstract. These “principles” sounded so detailed as to miss important mutation types that happened not to occur in this system. For example, the evolution of antibiotic resistance through horizontal gene transfer, or evolution via chromosome duplication through chromosomal mis-segregation. Maybe, the general principle is what the authors wrote in their Discussion: the observed frequency of an evolutionary strategy is determined by the probability of the particular type of mutation and the fitness advantage conferred by such a mutation. Thus, loss-of-function mutations in non-essential genes will be far more common than reduction-of-function mutations in essential genes, which will be more common than gain-of-function mutations (unless it is through chromosome duplication or horizontal gene transfer, which can occur at a high frequency).

2) In Figure 4, the relative advantage of a mutant over SM does not translate quantitatively to the relative advantage of the mutant over WspF S301R. Please discuss this.

3) In the subsection headed “Fitness of double and triple mutants”, the authors state: “In the invasion assay all mutants, except the PFLU0458 E45*, invaded the SM population successfully (Figure 5).” However according to Figure 5, the fitness of PFLU0458 E45* is similar to that of PFLU4443 PFLU4414 or PFLU4443 E398*. Thus, this seems like a contradiction.

4) It would be a nice addition to the paper if there were a summary figure or discussion that attempted to put everything that they now know about this system on a common scale of mutational rarity. If they observed 1,000,000 WS mutants, how many would they expect to be in each of the dominant and these lesser categories? (At least to a first approximation.) It also would be helpful to color or symbol code the different mutations by type so that this aspect could be more easily followed from Figure 3 to Figures 4 and 5.

5) The presentation of the paper could be improved with a big table. As it stands, the paper falls into a presentation mode that is common when there are many data to summarize: a shopping list of case studies is described in the text, one by one. The conceptual framework of the presentation in this manuscript is clear, but there is too much detail to follow. I thus suggest that a big summary table be created that groups the mutations under the different classifications that are described. The table can include subcategories, details of the type of mutation, the gene affected, etc. But it should enable the reader to see everything at a glance. Then, if some cases require further explanation, the text can do that. But getting the shopping list out of the text will allow the paper to focus on the points of general interest. Incidentally, that kind of table is the very type that could make it into a textbook, a worthy goal of any paper. (If need be, perhaps the data could be broken into two tables.)

---

## [Author Response]

*Lind et al. use the* Pseudomonas fluorescens *unshaken microcosm experimental evolution system to deeply profile different mutations that can give rise to the wrinkly spreader phenotype. Neither the trick of knocking out one pathway to see how evolution can still achieve a given phenotype nor the target-size argument for predicting evolutionary pathways are especially new. For example, the Barry Hall studies of evolved beta galactosidase used the former approach. There are also multiple examples of profiling the steps in evolving new metabolic capabilities in the 1984 Mortlock* “*Microorganisms as model systems for studying evolution*” *volume that follow the* ‘*target-size*’ *paradigm of a regulatory mutation knocking out a repressor first to achieve overexpression, followed by rarer refinement mutations at specific sites in an enzyme that change its specificity*.

We acknowledge that the work has some semblance to that of Barry Hall and others. Indeed, numerous studies have exploited mutational compensation for defective phenotypes as a way of obtaining insight into the genetic and phenotypic bases of traits. However, there are important differences between these studies and the work presented in our paper.

Firstly, we do not remove the biochemical activity required for the phenotype, but instead successively peel away regulatory pathways that stand to activate the structural component (in our case the *wss* operon required for the WS phenotype) upon mutation. In our work, removing commonly traversed mutational pathways does not have a deleterious effect on fitness: this cannot be said for the beta galactosidase case where a new biochemical function must evolve to compensate for a lost function.

Secondly, there is a difference in motivation between the standard “compensatory mutation” experiment and our work. For the most part others have been interested in compensatory evolution (underpinned by spontaneous mutation, or in vitro mutagenesis) as a means of restoring a lost function in the hope that new knowledge functional knowledge will emerge. For us the motivation comes from recognition that the standard explanation for parallel evolution (strong selection) is but one of a number of possible explanations for the repeated occurrence of the same phenotypic and even genotypic solutions to a common selective challenge. The problem, as others and we have pointed out, is to come up with an experimental model along with a sensible design to rigorously address the question as to the number of pathways that evolution (in a given context) can follow. This in and of itself is no trivial matter. To the best of our knowledge we have performed a unique experimental analysis and one that we hope will interest the evo-devo community, as much as to the field of experimental microbial population biology.

*1) I am not sure about the* “*general principles*” *quoted in the Abstract. These* “*principles*” *sounded so detailed as to miss important mutation types that happened not to occur in this system. For example, the evolution of antibiotic resistance through horizontal gene transfer, or evolution via chromosome duplication through chromosomal mis-segregation. Maybe, the general principle is what the authors wrote in their Discussion: the observed frequency of an evolutionary strategy is determined by the probability of the particular type of mutation and the fitness advantage conferred by such a mutation. Thus, loss-of-function mutations in non-essential genes will be far more common than reduction-of-function mutations in essential genes, which will be more common than gain-of-function mutations (unless it is through chromosome duplication or horizontal gene transfer, which can occur at a high frequency)*.

We have changed the Abstract and the Discussion to make it clear that the principles specifically apply to evolutionary challenges that can be solved by gene activation.

*2) In*
Figure 4*, the relative advantage of a mutant over SM does not translate quantitatively to the relative advantage of the mutant over WspF S301R. Please discuss this.*

We have added the following sentence to the section where the fitness assays are described: “Thus the two assays focus on different aspects of the adaptive radiation that are not necessarily directly correlated, for example a mutant can be superior in initial attachment, but grow slowly at the air-liquid interface. In addition, fitness in static microcosms is frequency-dependent (45).”

*3) In the subsection headed “Fitness of double and triple mutants”, the authors state:* “*In the invasion assay all mutants, except the PFLU0458 E45*, invaded the SM population successfully (*Figure 5*).*” *However according to*
Figure 5*, the fitness of PFLU0458 E45* is similar to that of PFLU4443 PFLU4414 or PFLU4443 E398*. Thus, this seems like a contradiction*.

After double-checking the data we confirmed the reviewers’ observation. The sentence has been changed to: “In the invasion assay all mutants, except the PFLU0458 E45*, PFLU4443 E398* and PFLU4443 PFLU4414, invaded the SM population successfully (Figure 5) (two tailed t-test p>0.01 vs. the control).”

*4) It would be a nice addition to the paper if there were a summary figure or discussion that attempted to put everything that they now know about this system on a common scale of mutational rarity. If they observed 1,000,000 WS mutants, how many would they expect to be in each of the dominant and these lesser categories? (At least to a first approximation.) It also would be helpful to color or symbol code the different mutations by type so that this aspect could be more easily followed from*
Figure 3
*to*
Figures 4 and 5.

We have added a figure (Figure 7) that schematically shows an estimation of the relative mutant frequencies. These estimates are a first approximation based on knowledge from previous published work and the data herein. In a forthcoming paper we will independently verify these frequencies and provide more precise figures. All figures/table are now color-coded consistently throughout the paper.

*5) The presentation of the paper could be improved with a big table. As it stands, the paper falls into a presentation mode that is common when there are many data to summarize: a shopping list of case studies is described in the text, one by one. The conceptual framework of the presentation in this manuscript is clear, but there is too much detail to follow. I thus suggest that a big summary table be created that groups the mutations under the different classifications that are described. The table can include subcategories, details of the type of mutation, the gene affected, etc. But it should enable the reader to see everything at a glance. Then, if some cases require further explanation, the text can do that. But getting the shopping list out of the text will allow the paper to focus on the points of general interest. Incidentally, that kind of table is the very type that could make it into a textbook, a worthy goal of any paper. (If need be, perhaps the data could be broken into two tables.*)

We have generated a table (Table 1) that summarises the mutational categories and presents the types of mutations that have been found. This has also been color-coded by mutational category. Even though we agree that there are plenty of details about the mutations in the text, we find it difficult to significantly reduce this without a negative impact on the credibility of the paper’s conclusions. Because the suggested classifications are based on interpretation of the mutational patterns (and not biochemical or biophysical data) it will not be possible for the reader to evaluate our claims without detailed descriptions and interpretations about the context of mutations in each gene. As this section is about 10% of the text in the article, we believe a further reduction would lead to a minor improvement in readability while seriously impacting the credibility of our conclusions.